# Microwave-Assisted Solvothermal Synthesis of Mo-Doped TiO_2_ with Exceptional Textural Properties and Superior Adsorption Kinetics

**DOI:** 10.3390/nano12122051

**Published:** 2022-06-15

**Authors:** Yahia H. Ahmad, Aymen S. Abu Hatab, Assem T. Mohamed, Mohammed S. Al-Kuwari, Amina S. Aljaber, Siham Y. Al-Qaradawi

**Affiliations:** 1Department of Chemistry and Earth Sciences, College of Arts and Sciences, Qatar University, Doha 2713, Qatar; yahiashoeb@qu.edu.qa (Y.H.A.); aymenabuhatab@hotmail.com (A.S.A.H.); asemtaha92@gmail.com (A.T.M.); a.s.aljaber@qu.edu.qa (A.S.A.); 2Department of Chemistry, Faculty of Science, Universiti Putra Malaysia UPM, Serdang 43400, Selangor, Malaysia; 3Ministry of Municipality and Environment, Doha 7634, Qatar; msakuwari@mm.gov.qa

**Keywords:** titanium oxide, Mo-doped TiO_2_, microwave-assisted synthesis, adsorption, Rhodamine B

## Abstract

Assigned to their outstanding physicochemical properties, TiO_2_-based materials have been studied in various applications. Herein, TiO_2_ doped with different Mo contents (Mo-TiO_2_) was synthesized via a microwave-assisted solvothermal approach. This was achieved using titanium (IV) butoxide and molybdenum (III) chloride as a precursor and dodecylamine as a surface directing agent. The uniform effective heating delivered by microwave heating reduced the reaction time to less than 30 min, representing several orders of magnitude lower than conventional heating methods. The average particle size ranged between 9.7 and 27.5 nm and it decreased with increasing the Mo content. Furthermore, Mo-TiO_2_ revealed mesoporous architectures with a high surface area ranging between 170 and 260 m^2^ g^−1^, which is superior compared to previously reported Mo-doped TiO_2_. The performance of Mo-TiO_2_ was evaluated towards the adsorption of Rhodamine B (RhB). In contrast to TiO_2_, which revealed negligible adsorption for RhB, Mo-doped samples depicted rapid adsorption for RhB, with a rate that increased with the increase in Mo content. Additionally, Mo-TiO_2_ expressed enhanced adsorption kinetics for RhB compared to state-of-the-art adsorbents. The introduced synthesis procedure holds a grand promise for the versatile synthesis of metal-doped TiO_2_ nanostructures with outstanding physicochemical properties.

## 1. Introduction

Extensive industrial activities are usually accompanied by the release of a wide range of hazardous materials that cause severe environmental pollution. In this context, different technologies have been removing these contaminants and mitigating their harmful effects on humans and ecosystems. Examples of these technologies are solvent extraction, chemical precipitation, coagulation, flocculation, membrane filtration, electrochemical oxidation, and photo-degradation. Among them, adsorption triggered great interest as an environmentally friendly technology that is credible for remediation of different types of pollutants. Compared to other techniques, adsorption has many advantages: ease of operation, environmental-friendliness, low operation cost, low energy input, and absence of toxic residues [1,2]. An ideal adsorbent should have enhanced physicochemical properties such as a high surface area, high adsorption capacity, and an enhanced chemical stability over a wide range of pH [3].

Over the last decades, TiO_2_ has been employed in a wide range of applications due to its unique merits such as high resistance to photo-corrosion, biocompatibility, remarkable chemical and thermal stabilities, non-toxicity, and low cost [4,5]. These applications extend from heterogeneous catalysis to energy storage, gas sensing, biomedical applications, food industry, and cosmetics [6,7,8,9,10,11,12,13]. TiO_2_ can be regarded as an ideal adsorbent owing to its non-toxic nature and its stability over a wide range of pH [14]. Accordingly, it was dedicated as an adsorbent for a wide variety of materials such as metal ions [15], dyes [16], gaseous molecules [17], and other organic compounds [18]. The physicochemical properties of TiO_2_ and its function as well can be tuned by several approaches. Among them, doping was adopted to control the electronic and the chemical properties of TiO_2_. In this context, different types of dopants were investigated to tolerate the properties of TiO_2_, such as metals, i.e., Fe, Mn, V, Cr, Cu, etc., and nonmetals, i.e., N, F, S, C, and P. Metal-doping has several merits such as low cost, ease of control, and a stable doped structure [19]. Doping of TiO_2_ with Mo triggered great interest, which aroused the radius of Mo^5+^ (0.61 Å) and Mo^6+^ (0.59 Å) and which are similar to that of Ti^4+^ (0.605 Å). This means that Mo^5+^/Mo^6+^ can effectively substitute Ti^4+^ in its lattice structure with low lattice distortion [20]. Doping of TiO_2_ with Mo cannot only influence the surface properties, but also affect the electronic and the optical properties and the material.

Several approaches have investigated the synthesis of Mo-doped TiO_2_, such as sol-gel, hydrothermal, spray pyrolysis, and magnetron sputtering [21,22,23,24]. For instance, Feng et al. studied the synthesis of Mo-doped TiO_2_ via a one-pot hydrothermal method at 473 K using dihydroxybis (ammonium lactato) titanium (IV) and ammonium heptamolybdate tetrahydrate as metals precursors [25]. The photocatalytic performance of the prepared materials was investigated for photocatalytic reduction of CO_2_ to methane. Similarly, Esposito et al. demonstrated the synthesis of Mo-doped TiO_2_ via a reverse micelle sol-gel approach using titanium butoxide and ammonium heptamolybdate tetrahydrate as metals precursors, cyclohexane as an oil phase, and polyoxyethylene (20) oleyl ether and Brij O20 as surfactants [26]. Their results exhibited that at Mo concentrations ≥ 2.5%, all phases of TiO_2_ coexist, i.e., anatase, rutile, and brookite. In addition, they found that the bandgap energy decreased with increasing the Mo content up to 7.5%, however, it increased at higher concentrations (10%) owing to the Moss−Burstein effect [26]. Compared to classical heating methods, microwave-assisted synthesis is characterized by several attractive features such as the absence of direct contact between the reactants and the energy source, which afford lower energy inputs, high heating rates, faster kinetics, better reaction yield, better control on the reaction parameters, and more reproducibility of products which are usually characterized by narrow particle size distribution [27,28]. It was investigated for the synthesis of a wide variety of compounds such as metals, alloys, metal oxides, and hybrid materials [29]. Notwithstanding, the synthesis of TiO_2_ and TiO_2_ composites via microwave-assisted routes has been frequently investigated [30,31,32], however, the fabrication of metal-doped TiO_2_ by microwave-assisted technique was rarely investigated [33,34]. In particular, microwave-assisted synthesis of Mo-TiO_2_ was not emphasized in the reported literature.

In the present study, we demonstrated the synthesis of Mo-doped TiO_2_ via a microwave-assisted solvothermal route. This was achieved via employment of titanium (IV) butoxide and molybdenum (III) chloride as precursors and dodecylamine as a surface directing agent. The atomic ratio of Mo in samples ranges between 0.9% and 3.1%. The morphological and the spectral properties of the as-prepared samples were examined by SEM-EDX, TEM, N_2_ physisorption, PXRD, XPS, and Raman spectroscopy. Besides, their performance as adsorbents for the uptake of Rhodamine B (RhB) from aqueous solutions was examined. It is noteworthy stating that the investigated synthesis procedure afforded Mo-doped TiO_2_ samples with superb specific surface area, which is at least two times greater than literature-reported values for Mo-TiO_2_. Besides, the adsorption kinetics is remarkably high compared to previously reported adsorbents. This can open the avenue toward the application of this developed synthesis procedure for manufacturing of a wide variety of TiO_2_-based materials with remarkable textural properties and outstanding catalytic activity.

## 2. Materials and Methods

### 2.1. Materials

Titanium (IV) butoxide (97%), molybdenum (III) chloride (99.95%), dodecylamine (98%), and ethanol (99.7%) were purchased from (Sigma-Aldrich Co., St. Louis, MO, USA). Isopropanol and hydrochloric acid (37%) were purchased from VWR Chemicals Co ((VWR International S.A.S., Fontenay-Sous-Bois, France). All chemicals were utilized as received without further purification.

### 2.2. Materials Synthesis

#### Synthesis of Mo-Doped TiO_2_ (Mo-TiO_2_) Nanoparticles

Mo-TiO_2_ was synthesized by a microwave-assisted procedure (Figure 1). In a typical synthesis, 5 mL of titanium (IV) butoxide was added to 30 mL isopropanol containing 5 mL 20% HCl under stirring. To the previous mixture, different amounts of MoCl_3_ were added to the solution with continuous stirring followed by the addition of 0.6 g of dodecylamine and stirring for a further 30 min. The mixture was placed into a microwave reactor (Anton Paar, Monowave 300 (Anton Paar GmbH, Graz, Austria), operating at a frequency of 2.45 GHz with a maximum power generation of 600 W. The reaction was allowed to react for 30 min at 160 °C. After natural cooling, the as-formed product was collected after several cycles of washing with absolute ethanol/centrifugation at 10,000 rpm. Finally, the product was dried in a vacuum overnight at 60 °C followed by calcination in air at 500 °C, 4 h using a ramping rate of 1 C min^−1^. The Mo contents were determined by energy-dispersive X-ray (EDX), they were found to be 0.00, 0.92, 1.45, 2.23, and 3.09 atomic%, and they were designated as TiO_2_, Mo-TiO_2_-0.9, Mo-TiO_2_-1.5, Mo-TiO_2_-2.2, and Mo-TiO_2_-3.1, respectively.

### 2.3. Characterization

The morphology of the as-synthesized materials was examined via transmission electron microscopy (TEM) via Tecnai TF20 microscope (FEI Company, Eindhoven, Netherlands) at an operating voltage of 200 kV. The crystal structure was examined by X-ray diffraction (XRD) via an X’Pert Phillips diffractometer (Phillips-PANalytical, Almelo, Netherlands) equipped with Cu-kα radiation (*λ* = 1.54059 Å). The electronic structures and oxidation states were investigated by X-ray photoelectron spectroscopy (XPS) with Axis Ultra DLD XPS (Kratos, Manchester, UK) equipped with a monochromatic Al-Kα radiation source (1486.6 eV). All binding energies were corrected against standard C 1 s peak, i.e., 284.6 eV. The textural properties were examined via N_2_ sorption experiments at liquid nitrogen temperature (77 K) using the Brunauer–Emmett–Teller (BET) method.

The zeta-potentials (ζ-potential) measurements were carried out using Zetasizer Nano ZSP instrument (Malvern Instruments Ltd., Worcestershire, UK) based on the electrophoretic mobility by applying Smoluchowski’s approximation. For each measurement, 5 milligrams of the sample were dispersed into 10 mL deionized water and sonicated for 10 min. After that, the pH value was adjusted by the addition of NaOH/HCl, and the steady state value was recorded.

### 2.4. Adsorption Activity

The performance of prepared materials as adsorbents was investigated towards the uptake of rhodamine B (RhB) as a model contaminant from aqueous solutions using the batch technique. In adsorption experiments, the amount of adsorbent was firstly fixed at 100 mg of the adsorbent in 100 mL of the dye solution, and the dye concentration was set at 10 ppm. Afterward, an adsorption study was performed at different concentrations between 15 ppm and 70 ppm. The adsorption process was studied at 20 °C. The concentration of RhB at different times was estimated from the calibration curves by measuring absorbance at *λ*_max_ = 554 nm, and the amount of adsorbed RhB (*q_t_*) was calculated as a function of time using the equation:(1)$$q_{t}=\frac{V\left(C_{o}-C_{t}\right)}{m}$$
where *V* is the volume of the RhB solution in liters, m is the mass of adsorbent in grams, and *C_o_* and *C_t_* are the initial and equilibrium concentrations of RhB at time *t*, respectively.

## 3. Results and Discussion

### 3.1. Synthesis of TiO_2_ and Mo-TiO_2_

TiO_2_ and Mo-TiO_2_ (of different Mo contents) were synthesized via microwave-assisted solvothermal method using titanium butoxide and MoCl_3_ as metals precursors, 2-propanol as a solvent, and dodecyl amine as a structure-directing agent. Intuitively, the synthesis of a material with a high surface area and inter/intraparticle porosity can be achieved by utilizing a hard or soft template during the preparation step. Different types of soft templates were employed for the synthesis of mesoporous TiO_2_-based materials such as nonionic surfactants such as Pluronic P123 [35] and Triton X-100 [36], in addition to ionic surfactants such as cetyl trimethyl ammonium bromide [37]. However, the employment of soft templates for the preparation of metal-doped TiO_2_ was not explored enough.

The formation model of Mo-TiO_2_ can be represented by Figure 2. Firstly, TBOT and MoCl_3_ are partially hydrolyzed by the acid to form oxide and/or hydroxide monomers, which may agglomerate in the form of small aggregates. The presence of surfactant molecules effectively disperses these aggregates by affording a cage-like environment that limits further nucleation and growth [38]. These small aggregates are assembled through electrostatic attraction between surfactant molecules and the formed nanoparticles. Induced by the rapid and high heating rate delivered by microwave irradiation, these nanoparticles undergo collisional growth and result in the formation of Mo-doped TiO_2_. Upon calcination, the surfactant molecules decompose and form porous structures with a high surface area (Figure 2).

### 3.2. Morphology

Figure 1 demonstrates the TEM images of as-prepared samples. Un-doped TiO_2_ is present as agglomerations of randomly-shaped nanoparticles. Mo-doped TiO_2_ depicted a greater extent of distortion in the shape with increasing the Mo content. This may be raised from decrease in the solubility of Mo in the TiO_2_ lattice, which resulted in the positioning of Mo species on the nanocrystallites or at the grain boundaries. Impressively, samples exhibit narrow particle size distributions, confirming the uniformity and the homogeneity of the nucleation and the growth processes throughout the reaction medium endowed by microwave irradiation. The calculated average particle sizes of samples are 27.5, 18.2, 13.5, 11.6, and 9.7 nm in the case of TiO_2_, Mo-TiO_2_-0.9, Mo-TiO_2_-1.5, Mo-TiO_2_-2.2, and Mo-TiO_2_-3.1, respectively, which reveals a decrease in the particle size with increasing the Mo-content, which is consistent with the previous studies. This may be attributed to the presence of Mo, which retards the crystal growth of TiO_2_ during the synthesis and is consistent with previous studies [25].

Figure 2 represents the TEM images and selected area electron diffraction (SAED) of TiO_2_-Mo-3.1 nanoparticles. The high-magnification TEM image (Figure 2a) showed clear lattice fringes. The SAED pattern demonstrates clear rings corresponding to the different planes of anatase structure and confirming the crystalline nature of the material. The HRTEM image (Figure 2c) showed lattice fringes with a d-spacing of 0.348 nm, which can be assigned to (101) plane of tetragonal anatase structure. The decrease in the value of d-spacing of Mo-TiO_2_ compared to the value of pure anatase (0.352 nm) can be attributed to the lattice distortion originated from substitution of Ti^4+^ with Mo^5+/6+^ (Appendix A).

### 3.3. Textural Properties

Figure 3 reveals the N_2_ adsorption–desorption isotherms and the corresponding pore size distributions of the investigated materials. All as-prepared materials exhibit a type IV isotherm with a hysteresis loop H5, which indicates the presence of opened and blocked pores [39]. The pore size distribution is remarkably an average radius ranging between 3.0 and 8.0 nm, implying the mesoporosity of the samples. It is noticeable that the capillary condensation took place over a wide range of P/P°, starting from 0.2 to 1.0. This affirms the non-uniformity of the pore sizes and the wide pore size distribution. The estimated values of surface area, pore volume, and pore size of studied samples are shown in Table 1. It is evident that the increase of Mo atomic ratio led to a remarkable increase in the surface area, which can be attributed to the decrease in the particle size observed from TEM analysis. The estimated values of surface areas and porosities are superior to previously reported values for Mo-TiO_2_ samples synthesized by other methods (Table 2), confirming the preferential textural properties delivered by the current approach.

### 3.4. XRD Analysis

Typically, TiO_2_ can exists in different forms i.e., anatase, rutile, brookite, and amorphous TiO_2_. Based on the preparation conditions, it can exist in a pure phase or a mixture of two or more phases. While rutile is stable at high temperatures, anatase is often the predominant phase upon preparation from solutions containing Ti precursor. XRD spectra were analyzed to investigate the crystalline structure of as-prepared nanomaterials. The XRD patterns of TiO_2_ and different compositions of Mo-TiO_2_ are manifested in Figure 4. Similar diffraction patterns were obtained for pure TiO_2_ and Mo-TiO_2_ samples with an absence of any diffractions ascribed to crystalline molybdenum oxides. This affirms either the complete integration of Mo in the crystal lattice of TiO_2_ or the amorphous nature of MoO_x_ species or their high dispersion, which hindered their XRD detection [26]. So far, the presence of molybdate species should be studied by Raman spectroscopy. The diffraction peaks at 2θ = 25.3°,37.8°, 48.1°, 53.9°, 55.1°, 62.7°, 68.8°, and 75.0° can be assigned to (101), (004), (200), (105), (211), (204), (116), and (215) planes of the anatase TiO_2_ (JCPDS Card No. 21-1272), respectively [40]. The diffraction peaks observed at 27.5°, 36.1°, and 69.0° can be indexed to the rutile phase (110), (101), and (301) planes, respectively (JCPDS Card No. 21–1276). The diffraction peak at 30.8° can be attributed to the (121) plane of brookite (JCPDS Card No. 29–1360). It should be noted that the intensities of rutile and brookite peaks remarkably diminish with increasing the Mo content, and the anatase became the predominant phase. The main peak (101) decreases and gets broader with increasing the Mo content, which can be attributed to a decrease in the crystalline nature of TiO_2_ and an increase in the lattice strain encountered by the size mismatching between Ti^4+^ and Mo^5+^/Mo^6+^ [21,41].

**Figure 4 nanomaterials-12-02051-f004:**
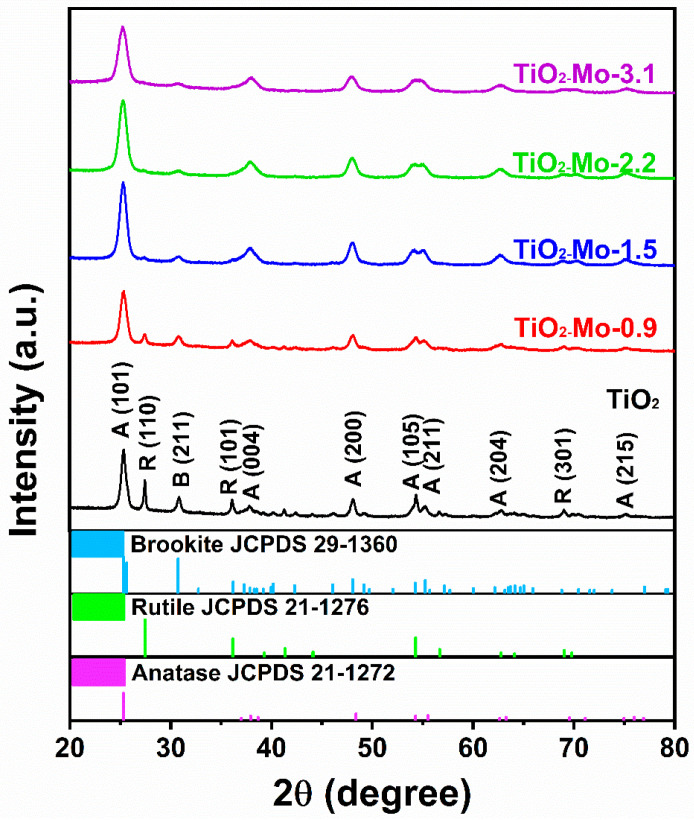
XRD spectra of studied nanomaterials.

**Table 2 nanomaterials-12-02051-t002:** Summary of some previous studies demonstrating the synthesis of Mo-doped TiO_2_.

Preparation Method	Metals Precursors	Reaction Conditions	Mo Ratio (%)	Specific Surface Area m^2^ g^−1^ (Pore Volume cm^3^ g^−1^)	Reference
Reverse micelle micro-emulsion sol-gel method	Titanium butoxide & ammonium heptamolybdate tetrahydrate	polyoxyethylene (20) oleyl ether (Brij O20) surfactants are dispersed in cyclohexane at 50 °C, then Ti precursor then emulsion broken by 2-propanol, dry, calcine at 500 °C, 2 h.	0 (TiO_2_) 1% 5% 10%	71 (0.091) 76 (0.112) 74 (0.141) 96 (0.137)	[42]
One-pot hydrothermal method	dihydroxy bis (ammonium lactato) titanium (IV) & ammonium heptamolybdate tetrahydrate	aqueous medium 200 °C, 12 h, 8 °C/min.	0 (TiO_2_) 0.1% 0.3% 0.5%	119 (0.32) 140 (0.38) 145 (0.40) 143 (0.36)	[25]
Sol-gel technique	Titanium butoxide & molybdenum pentachloride	Ethanol/acetylacetone/HCl sol is formed, left for 48 h, dry at 80 °C, then calcine at 450 °C	-	-	[43]
Hydrothermal method	TiO_2_ powder & ammonium heptamolybdate tetrahydrate	10 M NaOH, 180 °C/24 h Calcination 500 °C/1 h	0 (TiO_2_) 3% 5% 10%	112 156 168 172	[44]
Evaporation Induced Self Assembly	Titanium butoxide & ammonium heptamolybdate tetrahydrate	Ethanol/HNO_3_, rotary evaporator/17 h, calcine at 300 °C/1 h, then calcine at 400 °C	0 (TiO_2_) 1% 2% 3%	144 151 161 163	[45]
Hydrothermal method	titanium tetra-isopropoxide & molybdenum pentachloride	Isopropanol/H_2_O, hydrothermal 150 °C/20 h, calcine 500 °C/4 h	0 (TiO_2_) 0.02% 0.08%	129 127 124	[46]
Reverse micelle sol-gel synthesis	Titanium butoxide & ammonium heptamolybdate tetrahydrate	Polyoxyethylene, oleyl ether, Brij O20 surfactants, cyclohexane/H_2_O, then emulsion broken by 2-propanol, calcine at 500 °C/2 h	0 (TiO_2_) 0.83% 3.2% 5.8% 7.7% 9.75%	71 42 92 74 65 96	[26]
Microwave-assisted solvothermal method	Titanium butoxide & molybdenum (III) chloride	Dodecylamine surfactant, isopropanol/20% HCl, 160 °C, 30 min.	0 (TiO_2_) 0.9% 1.5% 2.2% 3.1%	91.7 (0.35) 173.7 (0.62) 191.7 (0.36) 206.6 (0.33) 256.6 (0.41)	This work

The crystallite sizes were calculated for all samples based on (101) diffraction peak using the Scherrer equation:(2)$$L=\frac{k\lambda }{\beta cos\theta }$$
where *L* is the crystallite size, *k* is constant (0.9), λ is the wavelength of X-ray radiation, and β is the full width at the half maximum of the peak. It can be observed that the crystallite size decreases with increasing the Mo content (Table 1). This can be explained on the basis that Mo slightly restrains the crystal growth, which is consistent with the previous studies [47]. This was explained on the basis that the existence of Mo into the lattice of TiO_2_ can constitute point defects that act as heterogeneous nucleation sites and hinder crystal growth [48]. The lattice strain, ε, was also evaluated from XRD data using the equation [21]:(3)$$\varepsilon =\frac{\beta cos\theta }{4}$$

It is noticeable that the lattice strain increasing with the increase of the Mo content (Table 1). This can be assigned to the lattice deformation induced by the difference in the size between Ti^4+^ and Mo^5+/6+^ [49], and is denoted as dopant-induced lattice strain [50].

### 3.5. Electronic Structure

The oxidation states of constituent elements were examined by XPS. The XPS survey spectra affirm the presence of Ti, O, and Mo in all Mo-doped samples (Appendix A). Figure 5a demonstrated the high-resolution spectrum of the Ti 2p core level. It reveals two peaks at about 464.7 eV and 458.7 eV, which can be indexed to the Ti 2p_1/2_ and Ti 2p_3/2_ of Ti^4+^ (TiO_2_), respectively. The small peak at 460.5 eV can be attributed to the Ti^3+^ present in the lattice of TiO_2_ [28]. The existence of Ti^3+^ species can be attributed to two main reasons. Firstly, the high temperature and pressures induced by microwave irradiation induce the formation of Ti^3+^ [51], which is evident from its existence in the pure TiO_2_ sample. In addition, Mo-doping enhances the conversion of Ti^4+^ to Ti^3+^ through intervalence charge transfer (IVCT) in which a charge compensation mechanism takes place via reduction of the more stable Ti^4+^ to the less stable Ti^3+^ together with oxidation of Mo^5+^ to the more stable Mo^6+^ [52].

The Mo 3d core level spectrum exhibited two peaks at approximately 232.8 and 235.9 eV, which are assigned to 3d_5/2_ and 3d_3/2_, respectively (Figure 5b). Deconvolution of spectra of Mo 3d show two peaks at 232.4 and 233.3 assigned to Mo^5+^ and Mo^6+^, respectively. Deconvolution of the O 1 s spectral region revealed the existence of three distinguishable peaks (Figure 5c). The first at 529.7 eV can be assigned to the lattice oxygen [53]. The other peak at 530.7 eV can be attributed to the oxygen at the oxygen-deficient regions [26], whereas the third peak at approximately 532.2 eV can be indexed to the surface hydroxyls and adsorbed oxygen species [54]. The binding energies of Ti2p_1/2_ and Ti2p_3/2_ are shifted to higher values with the insertion of Mo into the TiO_2_ lattice (Appendix A). This can be attributed to the higher electronegativity of Mo compared to Ti. So far, the substitution of Ti with Mo^5+/6+^ leads to a decrease in the electron density of Ti, and it shifts its binding energies to higher values [53,54]. More detailed XPS data are given in Appendix A.

Undoubtedly, the XPS analysis represents an indication to the surface composition of materials, and it can be a referent to the differences between the bulk and the surface concentrations. The Mo concentrations obtained from XPS measurements revealed a surface enrichment with Mo species, which may be attributed showing the formation of surface polymolybdates, especially at a higher Mo loading (see Table 1) [26].

### 3.6. Raman Spectroscopy

Raman spectra of the studied TiO_2_-based nanomaterials are demonstrated in Figure 6a. The Raman modes Eg, B1g, and A1g aroused from symmetric stretching of O–Ti–O, symmetric bending of O–Ti–O, and anti-symmetric bending of O–Ti–O vibrations, respectively [55]. All spectra reveal the Raman modes of anatase at 147 (Eg), 199 (Eg), 399 (B1g), 519 (A1g), and 639 (Eg) cm^−1^. The low-intensity peak at approximately 235 cm^−1^ can be attributed to two-phonon scattering aroused from the rutile phase [56]. It is noteworthy that MoO_3_ species have characteristic Raman peaks at 290, 667, 819, and 995 cm^−1^ [57]; intriguingly, doped-TiO_2_ samples revealed the Raman peaks of anatase, however, with a remarkable reduction in the peak intensities owing to the doping of anatase with Mo, which is consistent with previous studies [23]. At a higher Mo content, i.e., more than 3.1 at.% Mo-doped TiO_2_ samples revealed two additional peaks: one at approximately 845 cm^−1^ can be indexed to Mo–O–Mo vibration, and the peak at 960 cm^−1^ can be attributed to Mo = O stretching in octahedrally coordinated Mo species such as Mo_7_O_24_^6+^ and Mo_8_O_20_^4-^ (see Appendix A) [58]. These results implied that excess molybdenum species (strong Lewis acids) are available at the surface of Mo-TiO_2_ species, owing to the limited solubility of Mo in TiO_2_ [57].

### 3.7. Zeta-Potential Measurements

The electrophoretic mobility of samples was investigated by measuring the ζ-potential in water at different pH values between 2 and 10. Results are indicated in Figure 6b. Bare TiO_2_ depicted a point of zero charge at pH~5.7. The incorporation of Mo into the TiO_2_ lattice remarkably shifts the PZC towards lower pH values, confirming the increase in the surface acidity at higher Mo concentrations. This can be explained on the basis that the surface of nanoparticles is enriched with Mo^+5/+6^ species, which have high Lewis acidity [42]. The shift in the ζ-potential towards negative values over a wide pH range can be attributed to the enrichment of the surface with Mo species. This is consistent with the XPS results, which revealed an increased Mo/Ti atomic ratio at the surface compared to the value estimated by EDX analysis.

### 3.8. Adsorption Study

The activity of the investigated materials were tested as adsorbents for the RhB as a model contaminant. RhB is a highly water-soluble cationic red dye of the xanthene group. It is widely used as a coloring agent in the textile industry and as a fluorescent tracer. It has depicted carcinogenicity, neurotoxicity, and chronic toxicity toward both humans and animals [59,60]. The adsorption efficiency was evaluated by measuring the change in the absorbance of a RhB solution at a wavelength of 554 nm as a function of time. The adsorption of RhB was studied to investigate the impact of Mo-doping on the adsorbability (Figure 7a). In the presence of pure TiO_2_, RhB showed a negligible adsorption with time. This can imply a weak interaction between TiO_2_ and the dye. However, the adsorption capacity significantly increased with increasing the Mo content, and it followed the order TiO_2_-Mo-3.1 > TiO_2_-Mo-2.2 > TiO_2_-Mo-1.5 > TiO_2_-Mo-0.9 > TiO_2_. This can be explained based on the zeta-potential of TiO_2_ and Mo-TiO_2_ samples, which is shifted toward more negative values with increasing the Mo-content. This can enhance the electrostatic attraction between the cationic RhB molecules and the negatively charged surface of the adsorbent. In addition, the increase in the surface area caused by increasing the concentration of Mo can also afford more surface active sites, which enhance the adsorption process.

The adsorption of different concentrations of RhB (ranging between 15 and 70 mg L^−1^) as a function of contact time on TiO_2_-Mo-3.1 is presented in Figure 7b. At all initial concentrations, the rate of adsorption of RhB is fast at the beginning of the adsorption process, however, it gradually decreased at higher contact times until it reached a steady state. This can be attributed to the partial decrease in the concentration of the RhB, which is regarded as the driving force to further adsorption, in addition to the decrease in the number of available active sites [61]. The required time to reach equilibrium increased from 5 to 120 min when the initial RhB concentration increased from 1 to 40 mg/L. It should be noted that the time required to reach equilibrium increases with increasing the initial concentration of RhB. Similarly, the adsorption capacity increased from 15 mg/g to 60 mg/g, when the initial concentration of RhB increased from 15 to 70 ppm. This can be explained on the basis of the higher concentration of RhB representing a high driving force for the diffusion of RhB molecules, which accelerate the mass transfer of RhB from the bulk of the solution to the active sites of the adsorbent, especially those located at the inner surface.

At the early stages of the adsorption process, a large number of surface active sites are not occupied, and they are available for the adsorption process. So far, the adsorption process is rapid at the beginning of adsorption. After that, the adsorption process is retarded owing to the decrease in the number of un-occupied sites available for adsorption. During this stage, a small part of the RhB molecules can overcome the diffusion and penetrate inside the pores. At higher contact times exceeding 2 h, no significant changes were observed in the RhB uptake, which indicates that the equilibration time is approximately 2 h. Figure 7c exhibits the removal efficiency of RhB as a function of the initial concentration. It affirms the enhanced adsorption ability of Mo-TiO_2_, which maintains high removal efficiency even at relatively high RhB concentrations, i.e., 79.3% at initial concentration of 70 mg L^−1^ (see Figure 7c).

#### 3.8.1. Adsorption Isotherms

The adsorption data was analyzed in order to select the most suitable isotherm equation representing the current adsorption process. This was selected based on the values of the regression coefficient (R^2^) of their linear relations. The tested isotherms are:

The Langmuir isotherm can be represented by the equation:(4)$$\frac{C_{e}}{q_{e}}=\frac{1}{Kq_{max}}+\frac{C_{e}}{q_{max}}$$
where, *C_e_* (mg/L) is the equilibrium concentration of the adsorbent into the liquid, *q_e_* is the equilibrium adsorption capacity (mg/g), *q_max_* is the maximum monolayer coverage capacity of adsorbent (mg/g), and *K* is the Langmuir adsorption constant (L/mg).

The Freundlich isotherm is given as:(5)$$logq_{e}=logK_{F}+\frac{1}{n}logC_{e}$$
where *K_F_* is a constant related to the sorption capacity and *n* is a constant representing the favorability of the sorption system.

The Temkin isotherm can be represented by the formula:(6)$$q_{e}=BlnA_{T}+BlnC_{e}$$
(7)$$B=\frac{RT}{b_{T}}$$
where, *b_T_* is the Temkin isotherm constant (J mol^−1^), *R* is the universal gas constant (8.314 J K^−1^ mol^−1^), T is the absolute temperature (K), *A_T_* is the Temkin isotherm equilibrium binding constant (L g^−1^), and *B* is the constant related to heat of adsorption.

The Dubinin–Radushkevich (D-R) isotherm can be expressed as:(8)$$\ln q_{e}=\ln q_{s}-\beta \varepsilon^{2}$$
(9)$$\varepsilon =RT\left(1+\frac{1}{C_{e}}\right)$$
where *q_s_* is the theoretical isotherm saturation capacity (mg g^−1^); *β* is the D-R isotherm constant with a dimension of energy, and *ε* is the Polanyi potential

The adsorption data was applied to the four isotherms (Figure 8) and the calculated constants are presented in Table 3. According to values of obtained regression coefficients, the Langmuir isotherm is the best model describing the adsorption data. The calculated maximum monolayer coverage capacity from the Langmuir isotherm was 69.01 mg g^−1^.

#### 3.8.2. Adsorption Kinetics

The adsorption mechanism and kinetics were studied by fitting of the experimental adsorption data to two kinetic models, i.e., pseudo-first order and pseudo-second order models. The fitness of both models was determined based on their R^2^ values of their correlation coefficients of their linear forms. The pseudo first-order kinetic model can be expressed as:(10)$$\log\left(q_{e}-q_{t}\right)=\log q_{e}-\frac{k_{1}t}{2.303}$$
where *k*_1_ represents the rate constant of the pseudo first-order adsorption (min^−1^). The linear form of pseudo second-order kinetic model can be presented by the equation:(11)$$\frac{t}{q_{t}}=\frac{1}{k_{2}q_{e}^{2}}+\left(\frac{1}{q_{e}}\right)t$$
where *k*_2_ represents the rate constant of the pseudo second-order adsorption (g mg^−1^ min^−1^). The plots representing the model are presented in Figure 9 and the calculated kinetic parameters are given in Table 4. At all concentrations, the calculated correlation coefficient (R^2^) for the pseudo second-order was much higher than obtained from the pseudo first-order. In addition, the values of $q_{e}$ estimated by the pseud-second order model is much closer to the experimental values compared to the pseudo-first order model. This implies more fitness delivered by the pseudo second-order model.

Based on the zeta-potential measurements, Mo-TiO_2_ exhibits a negative charge at pH 7, while Rh.B. has a positive charge at the same pH. Based on this, the adsorption process is induced by electrostatic interactions between the negative charges of Mo-TiO_2_ and the positive charge of Rh.B. Following this step, the adsorbed species undergo intra-particle diffusion through the pores to the internal surface. Finally, the adsorbent–adsorbate equilibrium is established at the interfacial region. It is noteworthy mentioning that the TiO_2_-Mo-3.1 sample expressed enhanced adsorption kinetics compared to other state-of-the-art adsorbents available in literature. This is evident by comparing the adsorption performance previously reported state-of-the-art and other commercial adsorbents (Table 5 and Appendix A).

The adsorption kinetics particle diffusion model was studied in order to get more insights on the adsorption mechanism and the rate-determining step. Generally, it postulates three steps for the adsorption process, which are the external mass transport, intra-particle diffusion, and adsorption at the interior surface site. The overall rate of adsorption may be controlled by one of the steps or a combination of more steps [62]. The rate-determining step (r.d.s.) is investigated by the plot of *q_t_* vs. t^1/2^. If one straight line passes through the origin results, this means that the intra-particle diffusion is the r.d.s. However, if multi-straight lines arise and none of them pass through the origin, this means that the intra-particle diffusion is not the only r.d.s. and a combination of many steps are involved in the adsorption [61]. Results imply that intra-particle diffusion is not the only controlling step and many steps are involved in the adsorption (Figure 10).

## 4. Conclusions

Nanosized crystalline TiO_2_ and Mo-doped TiO_2_ of different Mo contents were synthesized via a microwave-assisted solvothermal approach. X-ray diffraction revealed that well-crystalline structures, where anatase is the prevailing phase with the abundance of rutile as a minor component, decreases with increasing the Mo content. The crystallite size decreases with increasing the concentration of Mo, which is accompanied by an increase in the strain owing to the lattice distortion originating from the size difference between Ti^4+^ and Mo^5+/6+^. XPS study depicted that molybdenum exists in two oxidation states, i.e., Mo^6+^ as a main component and Mo^5+^ as a minor, while titanium is present as Ti^4+^ with a minority of Ti^3+^ species. The performance of investigated materials as adsorbents for RhB from aqueous solutions was studied. Results revealed that contrary to un-doped TiO_2_, which showed negligible adsorbability, adsorption is enhanced upon increasing the Mo content. The adsorption data was fitted to different isotherms and kinetic models, and it was found to best fit the Langmuir adsorption isotherm and the pseudo-second order model. Despite the complexity of the adsorption process, which can be influenced by many factors, the Mo-content was found to be the dominant parameter, where its increase led to an increase in the negativity of the ζ-potential and so far enhances the adsorption of cationic RhB species. For clarity, the introduced materials demonstrated outstanding textural properties and enhanced performance as adsorbents. This may open the avenue toward synthesis of analogous TiO_2_-based nanomaterials that can be employed in environmental remediation.

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
