# Peer review of "Microwave-Assisted Solvothermal Synthesis of Mo-Doped TiO2 with Exceptional Textural Properties and Superior Adsorption Kinetics"

_nanomaterials, 2022, doi:10.3390/nano12122051_

Round 1

Reviewer 1 Report

This manuscript reports TiO2 doped with different Mo contents (Mo-14 TiO2) synthesized via microwave-assisted solvothermal approach, using titanium (IV) butoxide and molybdenum (III) chloride as metals precursors, and dodecylamine as a surface directing agent. Mo-TiO2 expressed enhanced adsorption kinetics for RhB compared to state-of-art adsorbents. I recommend its publication after minor revisions according to the following comments.

1.      HRTEM images of all samples should be provided in this work.

2.      XPS high-resolution spectra of all samples should be provided in this work.

3.      Ti3+ species could be determined by EPR spectra.

4.      The in-depth mechanism of superior adsorption kinetics should be provided in this work.

Author Response

Comment: Moderate English changes required.

Answer: The English editing has been improved throughout the manuscript.

Comment 1.      HRTEM images of all samples should be provided in this work.

Answer: HRTEM images for TiO2 and other samples were added to the supplementary part.

Comment 2.      XPS high-resolution spectra of all samples should be provided in this work.

Answer: XPS high-resolution spectra of all samples were provided in the supplementary part.

Comment 3.      Ti3+ species could be determined by EPR spectra.

Answer: Unfortunately, the EPR spectrometer is not available, however, the existence of Ti3+ species can be attributed to two main reasons. Firstly, the high temperature and pressures induced by microwave irradiation induce the formation of Ti3+ (ACS Appl. Mater. Interfaces 2014, 6, 691−696), which is evident from its existence in pure TiO2 sample. In addition, The Mo-doping enhances the conversion of Ti4+ to Ti3+ through intervalence charge transfer (IVCT) in which, a charge compensation mechanism takes place via reduction of the more stable Ti4+ to the less stable Ti3+ together with oxidation of Mo5+ to the more stable Mo6+ (J. Mater. Sci., 2019, 54, 5266-5279; Inorg. Chem., 2018, 57, 7279-7289).

Comment 4.      The in-depth mechanism of superior adsorption kinetics should be provided in this work.

Answer: Further discussion of the mechanism of adsorption was added to the discussion. Based on the zeta-potential measurements, Mo-TiO2 exhibits a negative charge at pH 7, while Rh.B. has a positive charge at the same pH. Based on this, the adsorption process is induced by electrostatic interactions between the negative charges of Mo-TiO2 and the positive charge of Rh.B. Following this step, the adsorbed species undergo intra-particle diffusion through the pores to the internal surface. Finally, adsorbent-adsorbate equilibrium is established at the interfacial region.  

Reviewer 2 Report

In this work, the authors synthesized nano crystalline TiO2 and Mo doped TiO2 via microwave-assisted solvothermal approach. The nanoparticles were characterized by SM-EDX, TEM, N2 physisorption, PXRD, XPS, and Raman spectroscopy. Authors further studied their adsorption performance of the uptake of Rhodamine B from aqueous solutions. In general, the experiments are well designed, data obtained looks very interesting, RhB absorption results demonstrate promising properties of Mo doped TiO2 nanoparticles. I would be happy to recommend this paper published in "Nanomaterials" as it is.

Author Response

Comment: English language and style are fine/minor spell check required

Answer: The English editing has been improved throughout the manuscript.

Reviewer 3 Report

1. Authors should state more clearly their opinion on the formation of single phase solid solutions in all or in several Mo-doped samples in study. What is known about the solubility limits of MoOx in the different polymorphs of TiO2? The most of the experiments in this study shows at the formation of homogeneous samples while several statements in the text imply the formation of inclusions of Mo oxides at the surface of TiO2 (lines 178-180 at p.5 etc.). A homogeneous distribution of dopant within a particle of solid solution is not mandatory. Hence, the surface of (Ti,Mo)O2 particles could be enriched with Mo, as it was found by the comparison of the results of XPS and EDX analyses. However, I haven’t seen any experimental evidence of the formation of the individual MoOx particles. Additional comments on this matter should be added to the Introduction.

2. The XPS data for the undoped TiO2 should be added for comparison. Due to the small amount of Mo and its tentatively non-homogeneous distribution in the particles, more detailed XPS study of all these samples could be not necessary. However, it is not clear, whether Ti3+ appears in TiO2 particles naturally, due to the features of their synthesis conditions, or its appearance is caused by the heterovalent Mo doping of TiO2. It could be revealed by the comparison of Ti 2p spectra of the doped and undoped TiO2.

 3. Fig. 4. The identification of XRD peaks would be very helpful; it especially concerns the peaks of rutile.

 4. Experimental. The power and frequency of the microwave generator should be given.

 5. Conclusion. “For clarity, the introduced materials demonstrated outstanding textural properties and enhanced performance as adsorbents.”

- If so, the enhanced performance of these materials should be confirmed by the direct comparison of their properties with similar industrially produced adsorbents or other valid reference samples.

Author Response

Comment: English language and style are fine/minor spell check required

Answer: The English editing has been improved throughout the manuscript.

Comment 1. Authors should state more clearly their opinion on the formation of single phase solid solutions in all or in several Mo-doped samples in study. What is known about the solubility limits of MoOx in the different polymorphs of TiO2? The most of the experiments in this study shows at the formation of homogeneous samples while several statements in the text imply the formation of inclusions of Mo oxides at the surface of TiO2 (lines 178-180 at p.5 etc.). A homogeneous distribution of dopant within a particle of solid solution is not mandatory. Hence, the surface of (Ti,Mo)O2 particles could be enriched with Mo, as it was found by the comparison of the results of XPS and EDX analyses. However, I haven’t seen any experimental evidence of the formation of the individual MoOx particles. Additional comments on this matter should be added to the Introduction.

Answer: Actually, there is a great disparity in the literature regarding to the solubility of Mo in TiO2 lattice. For example, some studies reported that it can exceed 10 at % (ACS Catal., 2016, 6, 6551-6559), while other studies reported that it is about 3 at. %. In our case, the presence of MoOx species cannot be identified by XRD owing to the possibility of their high dispersion or amorphous nature. However, they can be identified by the Raman spectrum. The absence of their characteristic peaks at 666, 820, and 995 cm-1 confirms the absence of MoOx species. Instead, some polymeric species are formed at higher concentrations of Mo (> 3.5 at. %) (Figure S9).

Comment 2. The XPS data for the undoped TiO2 should be added for comparison. Due to the small amount of Mo and its tentatively non-homogeneous distribution in the particles, more detailed XPS study of all these samples could be not necessary. However, it is not clear, whether Ti3+ appears in TiO2 particles naturally, due to the features of their synthesis conditions, or its appearance is caused by the heterovalent Mo doping of TiO2. It could be revealed by the comparison of Ti 2p spectra of the doped and undoped TiO2.

Answer:

The XPS data of pure TiO2 were added to the supplementary part for comparison and a discussion was added to account for the formation of Ti3+:

The existence of Ti3+ species can be attributed to two main reasons. Firstly, the high temperature and pressures induced by microwave irradiation induce the formation of Ti3+ (ACS Appl. Mater. Interfaces 2014, 6, 691−696), which is evident from its existence in pure TiO2 sample. In addition, The Mo-doping enhances the conversion of Ti4+ to Ti3+ through intervalence charge transfer (IVCT) in which, a charge compensation mechanism takes place via reduction of the more stable Ti4+ to the less stable Ti3+ together with oxidation of Mo5+ to the more stable Mo6+ (J. Mater. Sci., 2019, 54, 5266-5279; Inorg. Chem., 2018, 57, 7279-7289).

 Comment 3. Fig. 4. The identification of XRD peaks would be very helpful; it especially concerns the peaks of rutile.

Answer:

The XRD peaks were identified and referenced in terms of the different TiO2 phases.

 Comment 4. Experimental. The power and frequency of the microwave generator should be given.

Answer:

The mixture was placed into a microwave reactor (Anton Paar, Monowave 300, Austria) operating at a frequency of 2.45 GHz with a maximum power generation of 600 W.

Comment 5. Conclusion. “For clarity, the introduced materials demonstrated outstanding textural properties and enhanced performance as adsorbents.”

- If so, the enhanced performance of these materials should be confirmed by the direct comparison of their properties with similar industrially produced adsorbents or other valid reference samples.

Answer:

The adsorption performance of TiO2-Mo-3.1 was compared to two reference materials i.e. TiO2 (P25) and multi-walled carbon nanotubes (MWCNT). The data is shown in Figure S10 in the supplementary information.